# Downregulation of *Mirlet7* miRNA family promotes Tc17 differentiation and emphysema via de-repression of RORγt

**Phillip A Erice[1,2], Xinyan Huang[2†], Matthew J Seasock[1,2], Matthew J Robertson[3], Hui-Ying Tung[4], Melissa A Perez-Negron[2], Shivani L Lotlikar[2], David B Corry[2,4,5], Farrah Kheradmand[4,5,6], Antony Rodriguez[2,5]***

[1]Immunology Graduate Program, Baylor College of Medicine, Houston, United States; [2]Department of Medicine, Immunology & Allergy Rheumatology, Baylor College of Medicine, Houston, United States; [3]Dan Duncan Comprehensive Cancer Center, Baylor College of Medicine, Houston, United States; [4]Department of Pathology and Immunology, Baylor College of Medicine, Houston, United States; [5]Center for Translational Research on Inflammatory Diseases, Michael E Debakey, Baylor College of Medicine, Houston, United States; [6]Department of Medicine, Section of Pulmonary and Critical Care, Baylor College of Medicine, Houston, United States

**\*For correspondence:**
antonyr@bcm.edu

**Present address:** [†]Department of Pulmonary and Critical Care Medicine, The First Affiliated Hospital of Sun Yat-sen University, Guangzhou, China

**Competing interest:** The authors declare that no competing interests exist.

**Abstract** Environmental air irritants including nanosized carbon black (nCB) can drive systemic inflammation, promoting chronic obstructive pulmonary disease (COPD) and emphysema development. The *let-7* microRNA (*Mirlet7* miRNA) family is associated with IL-17-driven T cell inflammation, a canonical signature of lung inflammation. Recent evidence suggests the *Mirlet7* family is downregulated in patients with COPD, however, whether this repression conveys a functional consequence on emphysema pathology has not been elucidated. Here, we show that overall expression of the *Mirlet7* clusters, *Mirlet7b/Mirlet7c2* and *Mirlet7a1/Mirlet7f1/Mirlet7d*, are reduced in the lungs and T cells of smokers with emphysema as well as in mice with cigarette smoke (CS)- or nCB-elicited emphysema. We demonstrate that loss of the *Mirlet7b/Mirlet7c2* cluster in T cells predisposed mice to exaggerated CS- or nCB-elicited emphysema. Furthermore, ablation of the *Mirlet7b/Mirlet7c2* cluster enhanced CD8⁺IL17a⁺ T cells (Tc17) formation in emphysema development in mice. Additionally, transgenic mice overexpressing *Mirlet7g* in T cells are resistant to Tc17 and CD4⁺IL17a⁺ T cells (Th17) development when exposed to nCB. Mechanistically, our findings reveal the master regulator of Tc17/Th17 differentiation, RAR-related orphan receptor gamma t (RORγt), as a direct target of *Mirlet7* in T cells. Overall, our findings shed light on the *Mirlet7*/RORγt axis with *Mirlet7* acting as a molecular brake in the generation of Tc17 cells and suggest a novel therapeutic approach for tempering the augmented IL-17-mediated response in emphysema.

## eLife assessment

This **important** study indicates a significant role for individual *let-7* miRNA clusters in regulating generation of Tc17 CD8 cells and emphysema severity in a mouse model. The authors provide **convincing** evidence for let-7-mediated repression of the transcription factor RORgt and consequent modulation of IL-17-producing CD8 T cells, with correlated data from human emphysema material, though some of the effective *let-7* clusters remain to be tested for the ability to modulate disease. The findings, which substantially advance the understanding of roles that *let-7* miRNA

clusters play in modulating both T cell responses and emphysematous lung disease, will be of interest to T cell and lung disease researchers.

## Introduction

Chronic obstructive pulmonary disease (COPD) ranks as the third leading cause of mortality and is projected to account for over a billion deaths by the end of the 21st century (*GBD Chronic Respiratory Disease Collaborators, 2020*; *Institute for Health Metrics and Evaluation, 2019*; *Laniado-Laborín, 2009*). Currently, there are no treatment options to reverse emphysema, the most clinically significant variant of COPD, which often is progressive despite smoking cessation (*Bhavani et al., 2015*; *Anthonisen et al., 2002*).

Inhalation of fine particulate matter smaller than 2.5 microns ($PM_{2.5}$) found in outdoor and indoor air pollution as well as tobacco smoke are risk factors for COPD development (*Adeloye et al., 2022*; *Eisner et al., 2010*; *Hu et al., 2010*). We have previously shown that nanosized carbon black (nCB), a noxious chemical constituent of $PM_{2.5}$ found in the lungs of smokers, activates macrophages and dendritic cells orchestrating a pathogenic T cell-dependent inflammatory response and emphysema in mice (*Lu et al., 2015*; *You et al., 2015*; *Shan et al., 2009*; *Chang et al., 2022*).

Research over the last decade has pointed to the importance of dysfunctional inflammatory T cells in human COPD lung tissue and animal models of emphysema (*Grumelli et al., 2004*; *Xu et al., 2012*; *Williams et al., 2021*). Aberrant T cells are implicated in impaired host defense, exaggerated inflammation, and loss of self-tolerance in COPD (*Williams et al., 2021*; *Chen et al., 2023*; *Hogg et al., 2004*; *Maeno et al., 2007*; *Xu et al., 2012*). In this regard, we and others have demonstrated the role and pathogenicity of activated IFN-γ and IL-17-secreting subsets of $CD4^+$ and $CD8^+$ T lymphocytes including Th1, Th17, and Tc1 cells in clinical isolates and in mice with COPD (*Lu et al., 2015*; *You et al., 2015*; *Shan et al., 2009*; *Lee et al., 2007*; *Kheradmand et al., 2023*). The IL-17-secreting Th17 cells are particularly important as they promote the destruction of lung epithelium and recruitment of macrophages and neutrophils which then release proteolytic enzymes such as matrix metalloproteinases involved in the degradation of the lung structural matrix (*Barnes, 2016*; *Hoenderdos and Condliffe, 2013*). We previously demonstrated that intranasal inhalation of nCB in mice is sufficient to induce emphysema by stimulating lung T cell activation by dendritic cells and macrophages. Moreover, we found that genetic ablation of IL-17A can attenuate nCB- or cigarette smoke (CS)-induced alveolar destruction and airway inflammation (*Shan et al., 2012*; *You et al., 2015*). More recently, IL-17A- and IL-17F-secreting $CD8^+$ T cell (Tc17) subpopulation has been shown to play a critical role in the pathogenesis of several autoimmune and inflammatory disorders (*Globig et al., 2022*; *Huber et al., 2013*; *Srenathan et al., 2016*).

Both Th17 and Tc17 require the fate-deterministic transcription factor RAR-related orphan receptor gamma t (RORγt, encoded by *Rorc*) for differentiation and production of IL-17A (*Ivanov et al., 2007*). RORγt is the best-studied positive transcriptional regulator of IL-17A and IL-17F (*Ivanov et al., 2006*). In accordance with the importance of IL-17A transcription, RORγt expression has also been reported to be elevated in COPD patients and in mouse models of COPD (*Chu et al., 2011*; *Li et al., 2015*). However, the upstream pathophysiologic mechanisms that contribute to the induction of RORγt and differentiation of Tc17 cells in COPD have not been well elucidated.

We previously reported that *Mir22* inhibits HDAC4, promoting antigen-presenting cell (APC) activation in the lungs and inducing Th17-mediated emphysema in response to CS or nCB in mice (*Lu et al., 2015*). Additional microRNAs (miRNAs) that control APC and/or T cell-driven IL-17A$^+$ inflammation have been identified by others including the *Mirlet7* family (*Yang, 2012*; *Angelou et al., 2019*). miRNA expression-based studies have shown frequent downregulation of members of the *Mirlet7* family, including *Mirlet7a*, *Mirlet7b*, *Mirlet7c*, *Mirlet7d*, *Mirlet7e*, and *Mirlet7f* in human emphysematous lung tissue and in murine models of emphysema, but the mechanism(s) of action remain ill-defined (*Christenson et al., 2013*; *Pottelberge et al., 2011*; *Conickx et al., 2017*; *Izzotti et al., 2009*). *Mirlet7* genes are encoded across eight loci either as single genes or as polycistronic clusters which have confounded their analysis in vivo (*Rodriguez et al., 2004*). Previous studies used *Lin28b* transgenic overexpression in T cells to block the maturation and processing of the *Mirlet7* family. They showed an inhibitory role of *Mirlet7* family in Th17-driven response in the murine model of

experimental autoimmune encephalomyelitis attributed in part to regulation of IL-1 receptor 1 and IL-23 receptor (*Angelou et al., 2019*).

Here, we found that *MIRLET7*, notably the *MIRLET7A3/MIRLET7B* and *MIRLET7A1/MIRLET7F1/MIRLET7D* clusters, are suppressed in the T cells isolated from lungs of emphysema patients. Consistently, the analogous murine *Mirlet7b/Mirlet7c2* and *Mirlet7a1/Mirlet7f1/Mirlet7d* clusters, respectively, were similarly downregulated in pre-clinical emphysema models. We engineered mouse models with the specific loss-of-function (LOF) mutations of the *Mirlet7b/Mirlet7c2* and *Mirlet7a1/Mirlet7f1/Mirlet7d* clusters, respectively, in T cells as well as an inducible *Mirlet7g* gain-of-function (GOF) model to determine the T cell-intrinsic role of *Mirlet7* miRNA in emphysema pathogenesis. Deletion of *Mirlet7* miRNA in T cells worsened alveolar damage elicited by inhalation of CS or nCB, and increased infiltration of immune cells in the airways, including IL-17-producing CD8$^+$ T (Tc17) cells. Mechanistically, we found that *Mirlet7* controls type 17 differentiation by directly targeting the lineage-determining transcription factor, RORγt. In support of this conclusion, *Mirlet7* GOF mice were resistant to nCB-mediated induction of RORγt and Tc17 responses. Thus, we show a previously unappreciated role for *Mirlet7* as a repressor of RORγt and a molecular brake to the IL-17-mediated T cell inflammation in emphysema.

## Results

### The *Mirlet7b/Mirlet7c2* and *Mirlet7a1/Mirlet7f1/Mirlet7d* clusters are downregulated in lungs and T cells in COPD

To explore the involvement of *Mirlet7* in emphysema, we scrutinized the genomic locations and transcriptional annotation of *Mirlet7* members frequently downregulated in lung T cells isolated from smoker's lungs as well as mouse models of emphysema. This combined approach showed close linkage and high conservation of two *Mirlet7* clusters encoded from long intergenic non-coding RNA (linc)-like precursors in humans and mice (*Figure 1A*). To shed light on whether these *Mirlet7* clusters are downregulated in patients with COPD, we analyzed a published (GSE57148) lung RNA-seq dataset obtained from COPD (*n* = 98) and control (*n* = 91) subjects (*Kim et al., 2015*). Our analysis identified significant downregulation of the *Mirlet7ahg* and *Mirlet7bhg* gene cluster transcripts in COPD compared to control subjects (*Figure 1B*). We carried out quantitative PCR (qPCR) detection of *MIRLET7A*, which is encoded by both clusters, in lung tissue samples of smokers with emphysema and non-emphysema controls, detecting significant downregulation of *MIRLET7A* in emphysema samples relative to controls (*Figure 1C*). Because *Mirlet7* has been shown to participate in IL-17$^+$ T cell responses (*Angelou et al., 2019*; *Guan et al., 2013*; *Newcomb et al., 2015*), we next sought to determine if the expression pattern of *Mirlet7ahg* and *Mirlet7bhg*-derived *Mirlet7* members are impaired in purified CD4$^+$ T cells from emphysematous lungs. In support of our original hypothesis, the CD4$^+$ T cell expression of *Mirlet7a*, *Mirlet7b*, *Mirlet7d*, and *Mirlet7f* were all inversely correlated with more severe emphysema distribution in the lungs as determined by Computed Tomography (CT) scan (*Figure 1D*).

Next, we elucidated *Mirlet7b/Mirlet7c2* and *Mirlet7a1/Mirlet7f1/Mirlet7d* cluster expression (herein referred to as *Mirlet7bc2* and *Mirlet7afd*, respectively) in murine models of CS- or nCB-induced emphysema, respectively (*Figure 1E*). Paralleling our observations in human COPD and emphysema, mice with CS- or nCB-induced emphysema exhibited reduced expression levels of *pri-Mirlet7b/c2* and *pri-Mirlet7a1/f1/d* transcripts in the lung and from isolated lung CD4$^+$ and CD8$^+$ T cells (*Figure 1F–H*). Collectively, our expression results indicate suppression of *Mirlet7bc2* and *Mirlet7afd* in the lung and T cells in human and pre-clinical models of emphysema.

### Conditional deletion of the *Mirlet7bc2* cluster in T cells enhances nCB- or CS-induced emphysema

To investigate the in vivo requirement of the *Mirlet7bc2* within T cells, we generated conditional ready floxed mice (*let7bc2$^{f/f}$*). We then crossed *let7bc2$^{f/f}$* mice with *CD4-Cre* mice to generate *let7bc2$^{f/f}$; CD4-Cre* LOF mice (*let7bc2$^{LOF}$*) (*Figure 2A*). This approach allowed us to conditionally delete *Mirlet7bc2* in all T cells derived from the CD4$^+$CD8$^+$ double-positive (DP) stage (*Lee et al., 2001*; *Shi and Petrie, 2012*). We confirmed that *let7bc2$^{LOF}$* mice exhibit robust conditional deletion of *Mirlet7bc2* in DP thymocytes and peripheral T cells (*Figure 2B* and data not shown). Our *let7bc2$^{LOF}$* adult mice were

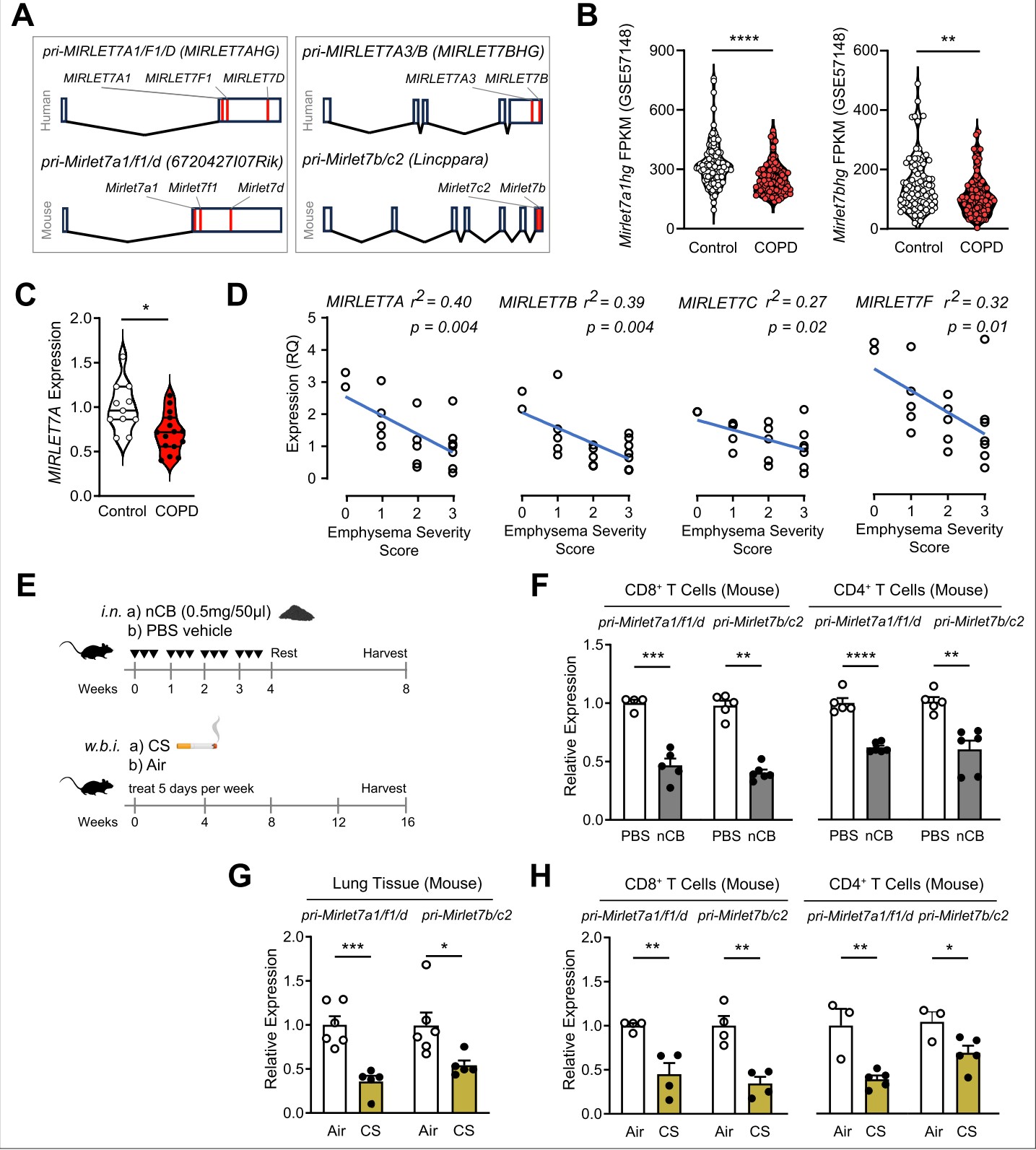

**Figure 1.** Repression of *Mirlet7* gene clusters in lung T cells from chronic obstructive pulmonary disease (COPD) patients and murine models of emphysema. (**A**) Schematic representation of the *Mirlet7* polycistronic transcripts in humans and in mice are shown. (**B**) Expression analysis of MIRLET7A1HG and MIRLET7BHG from the publicly available lung transcriptome dataset from RNA-seq of COPD and control patients (GEO: GSE57148). (**C**) Quantitative RT-PCR (qPCR) of mature MIRLET7A from resected lung tissue of COPD (*n* = 15) and control subjects (*n* = 11). (**D**) qPCR and regression

*Figure 1 continued*

analysis of MIRLET7A, MIRLET7B, MIRLET7D, and MIRLET7F expression to emphysema severity score based on CT: 0 = no, 1 = upper lobes only, 2 = upper/middle lobes, 3 = extensive pan lobular emphysema (*n* = 19). (**E**) Schematic diagram of experimental emphysema in mice induced by either intranasal (i.n.) instillation of nanosized carbon black (nCB) or exposure to cigarette smoke (CS) by whole-body inhalation (w.b.i.). qPCR analysis for *pri-Mirlet7a1/f1/d* and *pri-Mirlet7b/c2* from lung tissue or lung-derived CD8$^+$ and CD4$^+$ T cells of mice with emphysema elicited by (**F**) nCB or (**G, H**) CS (*n* = 3–6 per group). Data are representative of three independent experiments displayed as mean ± standard error of the mean (SEM). Mann–Whitney (**B, C**) or Student's *t*-test (**F–H**). *p < 0.05, **p < 0.01, ***p < 0.001, ****p < 0.0001.

The online version of this article includes the following source data for figure 1:

**Source data 1.** Source data for *Figure 1B–D, F–H*.

born at the expected Mendelian frequency and did not show any overt histopathologic or inflammatory changes in lungs histopathology up to 1 year of age in comparison to *let7bc2$^{f/f}$* control mice (*Figure 2—figure supplement 1A–D*). Furthermore, quantification of major immune populations and T cell subsets by flow cytometry in *let7bc2$^{LOF}$* were comparable to control mice under baseline conditions and with moderate aging (*Figure 2—figure supplement 1C, D*).

We next exposed *let7bc2$^{LOF}$* and *let7bc2$^{f/f}$* control mice to nCB or CS and examined the lungs under the context of experimental emphysema. Histomorphometry measurements of mean linear intercept (MLI) from hematoxylin and eosin (H&E)-stained sections revealed that the enlargement of alveolar spaces sustained from either nCB or CS exposure was exaggerated in *let7bc2$^{LOF}$* mice relative to controls (*Figure 2C–E*). Chronic inflammation in emphysema is characterized by the recruitment of macrophages and neutrophils to the lung tissue and airways (*Peleman et al., 1999*; *Senior and Anthonisen, 1998*). Internally consistent with MLI measurements, *let7bc2$^{LOF}$* mice treated with nCB showed significantly increased airway infiltration of macrophages and neutrophils in BAL fluid as compared to wild-type control animals (*Figure 2F*). Concomitant with these findings, expression levels of *Mmp9* and *Mmp12*, which are secreted by macrophages and neutrophils to degrade elastin and mediate alveolar damage, were elevated in airways of *let7bc2$^{LOF}$* mice exposed to CS versus controls (*Figure 2G*). As expected, *let7bc2$^{LOF}$* mice treated with nCB exhibit significantly less *pri-Mirlet7b/c2* transcript expression in isolated lung T cells relative to wild-type control mice (*Figure 2—figure supplement 2A* and data not shown). Collectively, our data suggest that the *Mirlet7bc2* cluster within T cells protects by dampening airway destruction and inflammation because the absence of this cluster worsens the severity of experimental emphysema in mice.

## The *Mirlet7b/Mirlet7c2* cluster negatively regulates T$_c$17 inflammation in emphysema

We sought to identify the T cell-intrinsic mechanisms that underlie the exaggerated inflammation observed in emphysematous *let7bc2$^{LOF}$* mice. We focused on the IL-17-mediated T cell response because it promotes neutrophil and macrophage recruitment in the lungs (*Beringer et al., 2016*; *Veldhoen, 2017*; *Shan et al., 2012*). Previously, we established the induction of CD4$^+$IL17$^+$ (Th17) cells along with CD4$^+$IFNγ$^+$ (Th1) cells in mice with chronic nCB exposure (*You et al., 2015*), however whether nCB similarly induces CD8$^+$IL17A$^+$ T cells (Tc17) or cytotoxic T cells (Tc1) had not been studied. The flow cytometric profiling of lung T cells revealed enriched proportions and counts of Tc1/Tc17 as well as Th1/Th17 cells in wild-type mice upon treatment with nCB (*Figure 3A, B* and *Figure 3—figure supplement 1A*). These findings suggests that nCB elicits both the type 17 and type 1 T cell responses, consistent with CS and elastase pre-clinical models of emphysema (*Zhang et al., 2019*).

We next interrogated the regulatory role of *Mirlet7bc2* in the type 17 and type 1 responses generated from exposure to nCB. Interestingly, *let7bc2$^{LOF}$* mice showed increased CD8$^+$IL17A$^+$ Tc17 cells relative to nCB control animals. In contrast, CD8$^+$IFNγ$^+$ and GZMA$^+$ Tc1 populations remained unperturbed with absence of *Mirlet7bc2*, suggestive of a more refined regulatory role on Tc17 differentiation (*Figure 3A, B*). There were no significant differences in either Th1 or Th17 cells when comparing nCB-treated *let7bc2$^{LOF}$* to wild-type controls, indicating the *Mirlet7bc2* was dispensable for their generation (*Figure 3C*). Regulatory T cells form a dynamic axis with Tc17/Th17 cells and act as a counterbalance to lung inflammation in emphysema (*Duan et al., 2016*; *Jin et al., 2014*). Therefore, we examined whether Tc17 cell alterations were driven by *Mirlet7bc2* acting on regulatory T cells (Tregs). The *let7bc2$^{LOF}$* mice showed no significant difference in the Tregs subset relative to controls in our model (*Figure 3D*). Together, our data support the notion that deletion of *Mirlet7bc2* is insufficient to

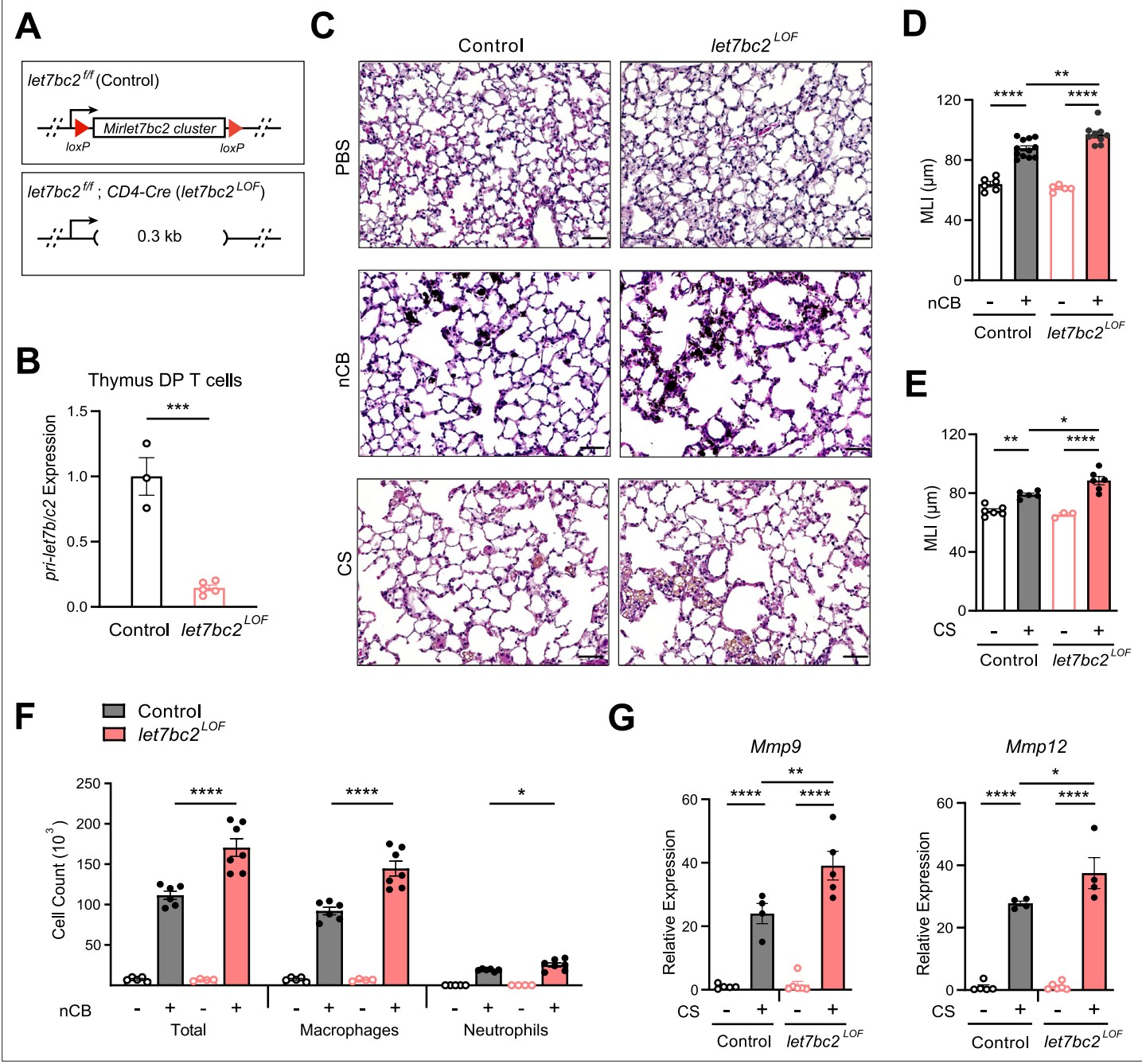

**Figure 2.** Deletion of the *Mirlet7bc2* cluster in T cells enhances nanosized carbon black (nCB)- or cigarette smoke (CS)-triggered emphysema. (**A**) Schematic representation of *let7bc2^LOF^* and control mice. (**B**) Quantitative PCR (qPCR) analysis of *pri-Mirlet7b/c2* from flow-sorted live, TCRβ⁺CD4⁺CD8⁺ double-positive (DP) thymocytes of control and *let7bc2^LOF^* mice (*n* = 3–5 per group). (**C–G**) Control and *let7bc2^LOF^* mice were exposed to phosphate buffered saline (PBS) or nCB over 4 weeks, or alternatively air or CS by whole-body inhalation of CS for 16 weeks. (**C**) Representative hematoxylin and eosin (H&E)-stained lung sections from PBS-, nCB-, or CS-exposed mice as indicated on each panel (×20 magnification; scale bars, 50 μm). (**D, E**) Mean linear intercept (MLI) measurement of lung morphometry. (**F**) Total and differential cell counts from bronchoalveolar lavage (BAL) fluid from controls and nCB-emphysemic mice (*n* = 4–7 per group). (**G**) *Mmp9* and *Mmp12* mRNA expression from BAL cells of air- and smoke-exposed control and *let7bc2^LOF^* mice (*n* = 4–6 per group). Data are representative of at least three independent experiments displayed as mean ± standard error of the mean (SEM) using Student's *t*-test (**B**) or two-way analysis of variance (ANOVA) with post hoc Tukey correction (**D–G**). *p < 0.05, **p < 0.01, ***p < 0.001, ****p < 0.0001.

The online version of this article includes the following source data and figure supplement(s) for figure 2:

**Source data 1.** Source data for *Figure 2B, D–G*.

*Figure 2 continued on next page*

*Figure 2 continued*

**Figure supplement 1.** T cell-specific deletion of the *Mirlet7bc2* does not promote lung inflammation or pathology with moderate aging.

**Figure supplement 2.** *Pri-Mirlet7b/c2* expression in lung CD8+ T cells of naive and nanosized carbon black (nCB)-exposed mice.

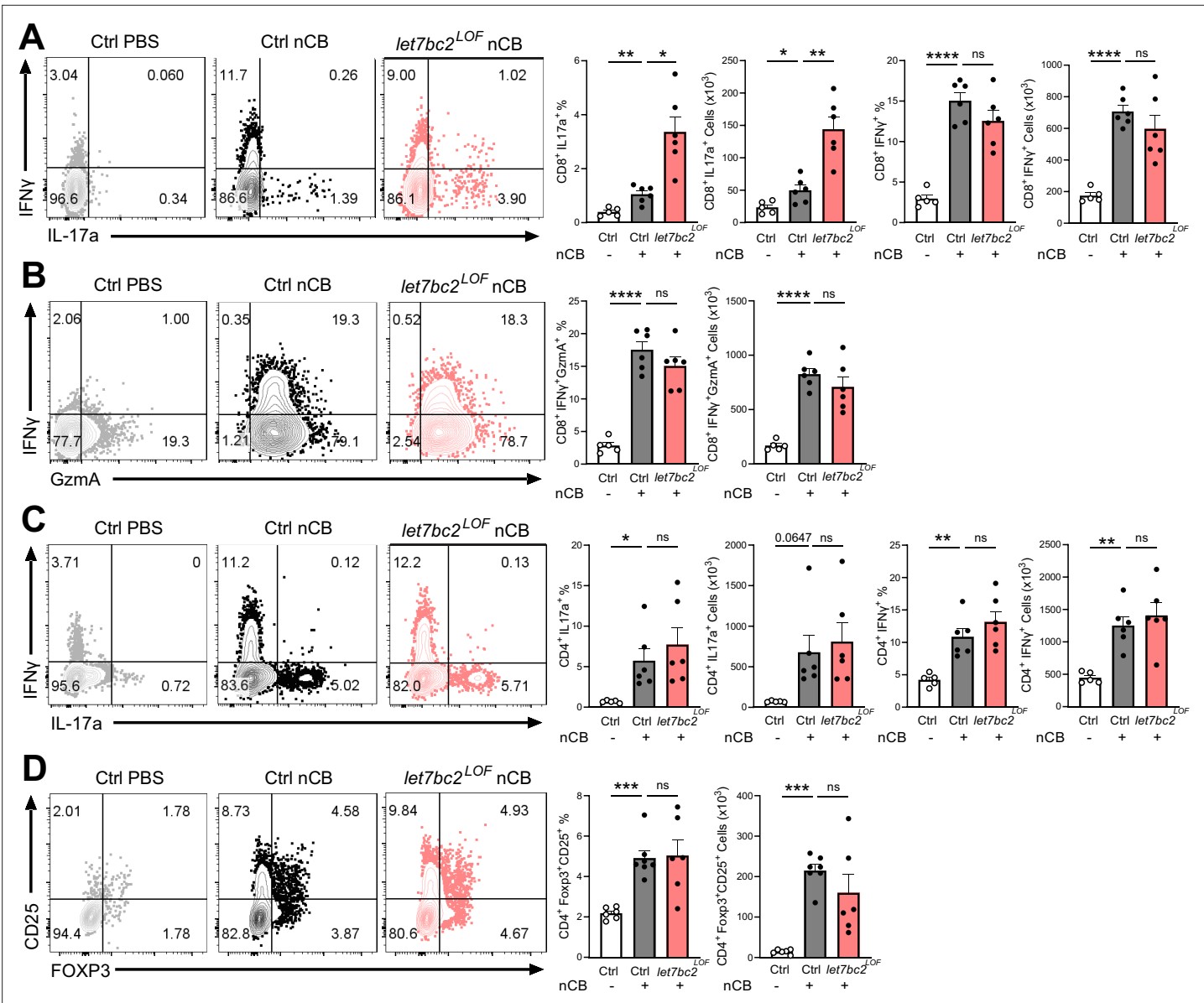

**Figure 3.** In vivo T cell ablation of the *Mirlet7bc2* cluster enhances Tc17 inflammatory response to nanosized carbon black (nCB) emphysema. Representative flow plots with percentage and counts of live TCRβ+ (**A**) CD8+IL-17a+ and CD8+IFNγ+, (**B**) CD8+IFNγ+GzmA+, (**C**) CD4+IL-17a+ and CD4+IFNγ+, and (**D**) CD4+FOXP3+CD25+ cells from the lungs of control (Ctrl) PBS vehicle- (*n* = 5–6), control nCB- (*n* = 6), and *let7bc2*LOF nCB-exposed mice. Data are representative of three independent experiments displayed as mean ± standard error of the mean (SEM) using analysis of variance (ANOVA) with post hoc Sidak correction. *p < 0.05, **p < 0.01, ***p < 0.001, ****p < 0.0001.

The online version of this article includes the following source data and figure supplement(s) for figure 3:

**Source data 1.** Source data for *Figure 3A–D*.

**Figure supplement 1.** T cell flow cytometry gating.

provoke Tc17 cell generation under homeostatic conditions. However, under the context of chronic inflammation in emphysema, the loss of the *Mirlet7bc2* cluster is intrinsic for the potentiation of T cells toward Tc17 differentiation.

## The *Mirlet7* family directly inhibits RORγt expression governing Tc17 differentiation in emphysema

We utilized the TargetScan predictive algorithm to identify putative *Mirlet7* targets that are known to control the IL-17-mediated T cell response (*Agarwal et al., 2015*). This analysis revealed that the 3′UTR region of *Rorc*, encoding RORγt, contains an evolutionarily conserved and complementary motif for the *Mirlet7* family (*Figure 4A*). Thus, we examined if *Mirlet7bc2* loss in T cells would stimulate and enhance RORγt. Initially, we carried out flow cytometric quantification for RORγt in thymocyte, splenic, and lung T cells of naïve control and *let7bc2^LOF^* mice up to 6 months of age. Our interrogation of RORγt mean fluorescent intensity (MFI) by flow cytometry showed induction of RORγt in single-positive CD8^+ and CD4^+ thymocytes, as well as peripheral splenic CD8^+ and CD4^+ T cells (*Figure 4B*). However, RORγt levels appeared unchanged in purified lung CD8^+ T cells and CD4^+ T cells of naive *let7bc2^LOF^* mice, alluding to a compensatory effect in homeostatic lung T cells (*Figure 4B*). Since we and others have shown that miRNAs are frequently associated with stress-dependent phenotypes, we posited that emphysematous *let7bc2^LOF^* T cells are poised toward induction of RORγt and production of IL-17^+ subsets after challenge with nCB. Indeed, nCB-emphysematous *let7bc2^LOF^* mice exhibited enhanced RORγt protein levels in both CD8^+ and CD4^+ T cells relative to control mice with emphysema (*Figure 4C*).

Because we had found that the *Mirlet7afd* cluster is downregulated in T cells isolated from COPD lungs in human and mice, and that the *Mirlet7* family operates with some functional redundancy, we generated mice with conditional deletion of the *Mirlet7afd* in T cells (*let7afd^LOF^*). The *let7afd^LOF^* mice aged up to 6 months did not exhibit overt lung histopathology and inflammatory changes (*Figure 4— figure supplement 1A–E*). Of particular interest, ablation of *Mirlet7afd* enhanced levels of RORγt in thymic and peripheral T cells of mice (*Figure 4D*). Overall, this indicates that independent *Mirlet7* clusters restrain RORγt expression levels from thymic development to peripheral T cells under homeostatic conditions. Next, we determined whether loss of *Mirlet7afd* in T cells likewise sensitizes mice toward induction of RORγt in nCB emphysema. Intranasal administration of nCB provoked increased RORγt expression in lung T cells of *let7afd^LOF^* mice compared to control mice (*Figure 4E*), supporting overlapping functionality between the *Mirlet7bc2* and *Mirlet7afd* in repression of RORγt within T cells.

To confirm that the *Mirlet7* family negatively regulates Tc17 cell differentiation, at least in part, cell autonomously in CD8^+ T cells, we purified naïve CD8^+ T cells from *let7bc2^LOF^* and control mice spleens and cultured these cells in vitro in the presence of Tc17 polarizing (TGFβ, IL-6, anti-IFNγ, IL-23, and IL-1β) or Tc1 polarizing (IL-2) conditions (*Flores-Santibáñez et al., 2018*). Our flow cytometric analysis confirmed the enhanced commitment of *Mirlet7bc2* deficient CD8^+ T cells toward Tc17 cells and IL-17A^+ production relative to control CD8^+ T cells (*Figure 5A, B*). Moreover, enhanced Tc17 cell differentiation mirrored the increased IL-17A detected in the supernatant from in vitro polarized cells as quantified by ELISA (*Figure 5C*). Parallel assessment of Tc1 differentiation did not detect a difference in CD8^+IFNγ^+ cells (*Figure 5A and D*). Altogether, these data recapitulated our in vivo findings that the *Mirlet7bc2* cluster negatively regulates Tc17 response but is dispensable in Tc1 cells. Finally, to determine whether Tc17 differentiation is likewise controlled by *Mirlet7afd*, we cultured naive CD8^+ splenocytes from *let7afd^LOF^* and controls under Tc17 conditions. As we had observed with *let7bc2^LOF^*, absence of *Mirlet7afd* in T cells further enhanced differentiation toward Tc17 cells as quantified by flow cytometry and ELISA (*Figure 5F, G*).

Next, we focused on *Rorc* as a potential direct target of *Mirlet7*, which could mechanistically mediate enhanced Tc17 differentiation in *let7bc2^LOF^* mice. Toward this objective, we tested whether *let7bc2^LOF^* or *let7afd^LOF^* naive CD8^+ T cells show elevated RORγt expression under either Tc0 or Tc17 differentiation conditions. In agreement with enhanced Tc17 differentiation, RORγt expression was differentially and significantly upregulated under both Tc0 and Tc17 differentiation conditions in *Mirlet7* LOF cells relative to controls (*Figure 5E, H*). To determine whether *Mirlet7* directly represses *Rorc* mRNA levels we cloned the 3′UTR of *Rorc* into luciferase constructs. These reporter assays with *Mirlet7b* expressing cells independently confirmed that *Mirlet7b* represses *Rorc* (*Figure 5I*, left). Furthermore, deletion of the putative *Mirlet7*-binding sequence (*Figure 4A*) abrogated repression by *Mirlet7b* (*Figure 5I*,

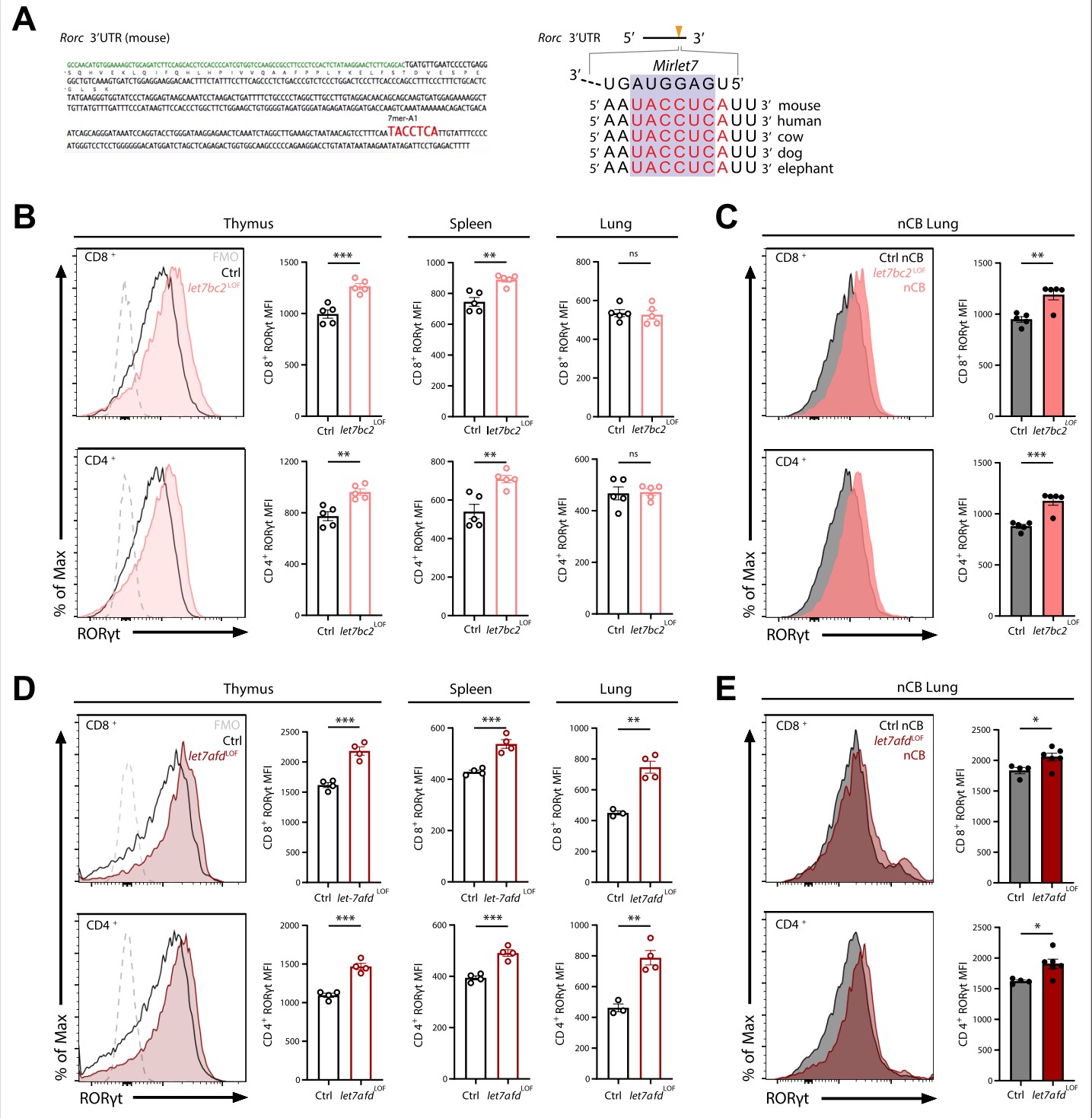

**Figure 4.** Deletion of either *Mirlet7bc2* or *Mirlet7afd* cluster in T cells enhances RORγt expression in vivo. (**A**) Left: Schematic representation of the murine *Rorc* 3'UTR with *Mirlet7*-binding site as identified by TargetScan. Right: Schematic of a conserved *Mirlet7* microRNA (miRNA) target sequence in the 3'UTR of *Rorc*. Flow analysis of RORγt expression by mean fluorescent intensity (MFI) quantification in live TCRβ⁺CD8⁺ or CD4⁺ T cells from indicated tissues of (**B**) naïve control (Ctrl) and *let7bc2*^LOF mice or (**C**) nanosized carbon black (nCB)-treated lungs by representative flow plot and MFI quantification (*n* = 5 per group). (**D**) RORγt expression by MFI quantification in naive *let7afd*^LOF mice thymus, spleen, and lungs (*n* = 3–4 per group), or (**E**) nCB-exposed lungs (*n* = 5 per group). Data are representative of at least three independent experiments displayed as mean ± standard error of the mean (SEM) using Student's *t*-test. *p < 0.05, **p < 0.01, ***p < 0.001.

The online version of this article includes the following source data and figure supplement(s) for figure 4:

*Figure 4 continued on next page*

*Figure 4 continued*

**Source data 1.** Source data for *Figure 4B–E*.

**Figure supplement 1.** T cell-specific deletion of the *Mirlet7afd* cluster does not promote lung inflammation or pathology with moderate aging.

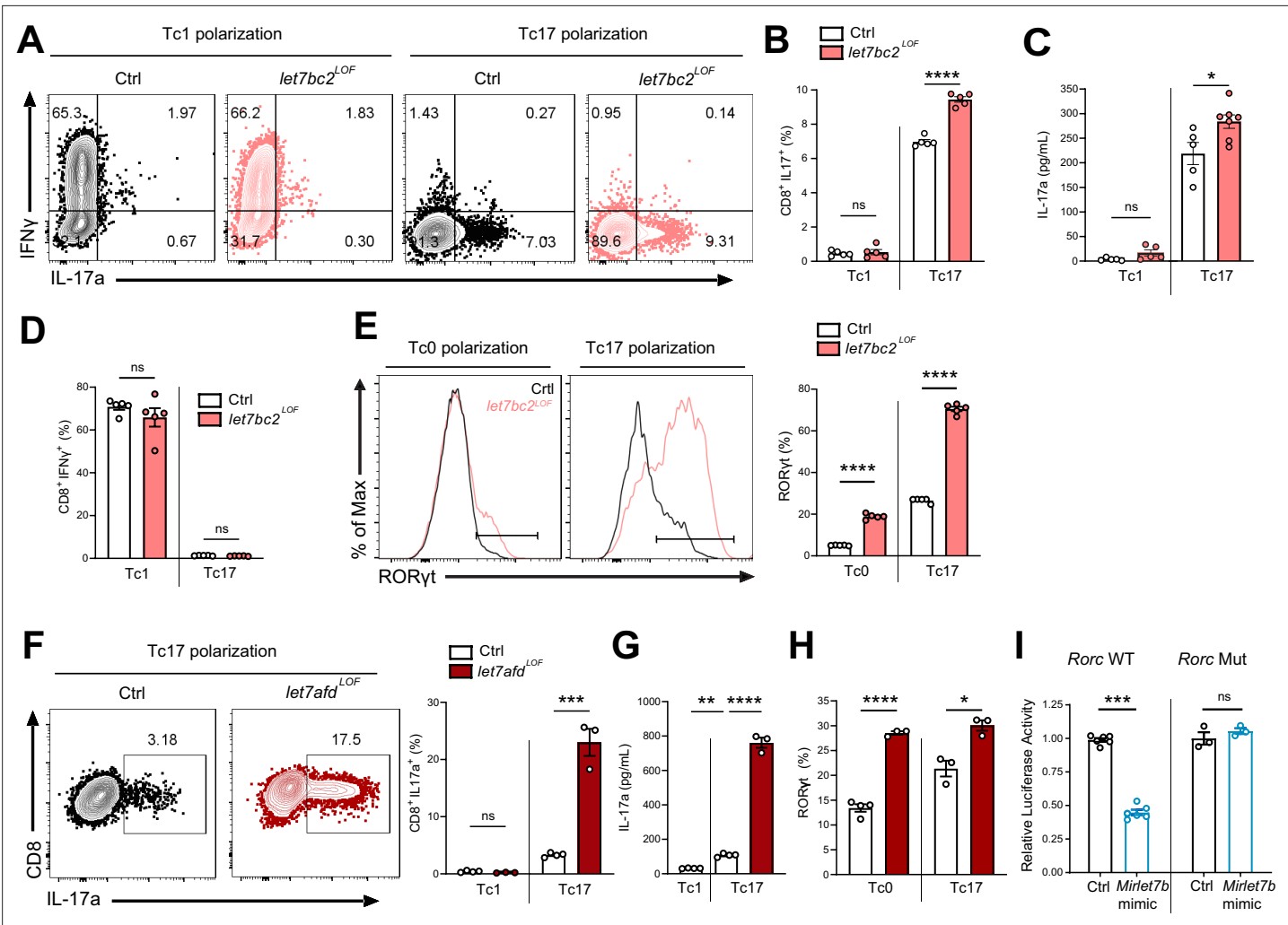

**Figure 5.** *Mirlet7* restricts Tc17 in vitro differentiation in part via direct targeting of *Rorc* mRNA. (**A**) Representative flow plots of live TCRβ⁺CD8⁺, IL-17a⁺, and IFNγ⁺ populations from Tc1 and Tc17 polarized naive splenic CD8⁺ T cells from control and *let7bc2^LOF* mice and (**B**) quantification of CD8⁺IL-17a⁺ cells (*n* = 5 per group). (**C**) ELISA of IL-17a from the supernatant of Tc1 and Tc17 polarized control and *let7bc2^LOF* cells (*n* = 5–6 per group). (**D**) Flow quantification of CD8⁺IFNγ⁺ populations in Tc1 and Tc17 polarized control and *let7bc2^LOF* cells (*n* = 5 per group). (**E**) Representative flow plot and quantification of RORγt from Tc0 or Tc17 differentiated naive splenic CD8⁺ T cells isolated from control and *let7bc2^LOF* mice (*n* = 5 per group). (**F**) Representative flow plots of CD8⁺IL-17a⁺ population frequency and quantification of Tc17 polarized naive splenic CD8⁺ cells of indicated mice polarized under Tc1 or Tc17 conditions. (**G**) ELISA of IL-17a from control, Tc1 (*n* = 4), control Tc17 (*n* = 4), and *let7afd^LOF* Tc17 (*n* = 3) polarized cells. (**H**) Quantification of RORγt from Tc0 or Tc17 in vitro polarized naive CD8⁺ T cells from control and *let7afd^LOF* mice (*n* = 3–4 per group). (**I**) Control (*Rorc* WT) or binding site mutant (*Rorc* Mut) 3'UTRs of *Rorc* were cloned downstream of the renilla luciferase reporter. Plasmids were cotransfected with either a control-miR (black bars) or *Mirlet7b* mimic (blue bars) duplex into cultured cells. Reporter activity was measured 24 hr after transfection and normalized to firefly activity. Data are representative of two (**H**), three independent experiments (**A–G**), or carried out in triplicate (**I**) and displayed as mean ± standard error of the mean (SEM) using Student's *t*-test. *p < 0.05, **p < 0.01, ***p < 0.001, ****p < 0.0001.

The online version of this article includes the following source data for figure 5:

**Source data 1.** Source data for *Figure 5B–I*.

right), thus confirming *Rorc* as a functional target. Overall, these in vitro experiments readily recapitulated an upstream regulatory role of *Mirlet7* in Tc17 differentiation, mediated in part, via direct suppression of RORγt.

## Enforced expression of *Mirlet7g* tempers RORγt T cell expression levels in experimental emphysema

To explore a potential protective role of *Mirlet7* in experimentally induced emphysema, we generated mice which allowed for selective induction of *Mirlet7g* in T cells using the published *rtTA-iLet7* mice crossed to CD4-Cre (herein referred to as *let7^GOF^*; *Figure 6A*; *Angelou et al., 2019*; *Belteki et al., 2005*; *Pobezinskaya et al., 2019*; *Wells et al., 2017*; *Zhu et al., 2011*). The rtTA-iLet7 mouse model has been utilized to promote ~two- to threefold rise in total *Mirlet7* activity in T cells (*Angelou et al., 2019*; *Wells et al., 2017*; *Wells et al., 2023*; *Angelou et al., 2019*). Steady-state *let7^GOF^* and control (*rtTA-iLet7*) mice were examined for compromised RORγt protein levels within thymocytes and peripheral T cells. Providing further evidence of *Mirlet7*-dependent regulation of *Rorc*, protein levels of RORγt were suppressed in CD8+ and CD4+ T cells of *let7^GOF^* mice relative to controls (*Figure 6B*). To determine whether enforced expression of *Mirlet7g* offered protection from experimental emphysema, *let7^GOF^* and control mice were treated with nCB and then examined for changes in lung pathology and T cell type 17 responses. The *let7^GOF^* mice did not exhibit any signs of lung inflammation or pathologic remodeling at baseline (*Figure 6C, D* and data not shown). Histopathologic analysis revealed a comparable degree of lung alveolar distension via morphometric measurements of MLI in nCB-treated *let7^GOF^* mice versus controls suggesting that enforced *Mirlet7g* expression is insufficient to protect the lung from emphysema (*Figure 6C, D*). On the other hand, evaluation of the IL-17+ response and RORγt levels in emphysematous lung T cells demonstrated that, in contrast to control nCB-treated mice, *let7^GOF^* mice exhibited dampened lung Tc17 and Th17 cell populations and were resistant to the induction of RORγt after nCB exposure (*Figure 6E, F*). Taken together, our *Mirlet7* LOF and GOF models demonstrate the necessity and sufficiency of *Mirlet7* to act as a molecular brake to the type 17T cell response through the direct regulation of RORγt, further our data suggest that nCB- or CS-mediated suppression of this braking mechanism furthers inflammation and exacerbates emphysema severity (*Figure 6G*).

## Discussion

miRNA expression-based studies of COPD patients and mice exposed to CS have reported downregulation of *Mirlet7* expression in lung tissues (*Conickx et al., 2017*; *Christenson et al., 2013*; *Schembri et al., 2009*). We and others explored the consequence of loss of *Mirlet7* expression/activity with synthetic oligonucleotides, sponges, lentiviral antisense knockdown, or via ectopic delivery of *Lin28b* (*Polikepahad et al., 2010*; *Viswanathan et al., 2008*; *Piskounova et al., 2011*), but studies pinpointing the role of individual *Mirlet7* clusters as potential drivers of lung inflammation and COPD within T cells remained elusive. In the present study, we established that the *Mirlet7* family members encoded by the *Mirlet7b/Mirlet7c2* and *Mirlet7a1/Mirlet7f1/Mirlet7d* clusters are downregulated in T cells from lungs of emphysema patients and emphysematous mice that were exposed to CS or nCB. Correspondingly, we demonstrated that in vivo genetic ablation of *Mirlet7b/Mirlet7c2* further sensitized mice to lung tissue destruction and emphysema upon treatment with nCB or CS. Mechanistically, our studies suggests that *Mirlet7b/Mirlet7c2* cluster prevents the emergence of CD8+ T cell differentiation into Tc17 cells during emphysema in part, by directly silencing of *Rorc*.

Tc17 cells are vital for defense against viral, fungal, and bacterial infections and they have also been associated with inflammation in various human diseases such as multiple sclerosis, inflammatory bowel disease, and cancer (*Huber et al., 2013*; *Globig et al., 2022*; *Corgnac et al., 2020*). In accordance with the potential pathogenic role of Tc17 cells as drivers of COPD, several studies detected increased cell numbers in airways and tissues of COPD patients as well as lungs of smoke-exposed animal models (*Chang et al., 2011*; *Zhou et al., 2020*; *Duan et al., 2013*). Other researchers also detected increased Tc17 subpopulations in tissues of COPD patients with infectious microbial exacerbations. In our earlier work to define the adaptive T cell immune responses in nCB-induced COPD, we predominantly focused on the pathogenic role of Th17 cells, but did not examine Tc17 cells (*You et al., 2015*). Here, we expand upon our prior observations, revealing that chronic exposure to nCB

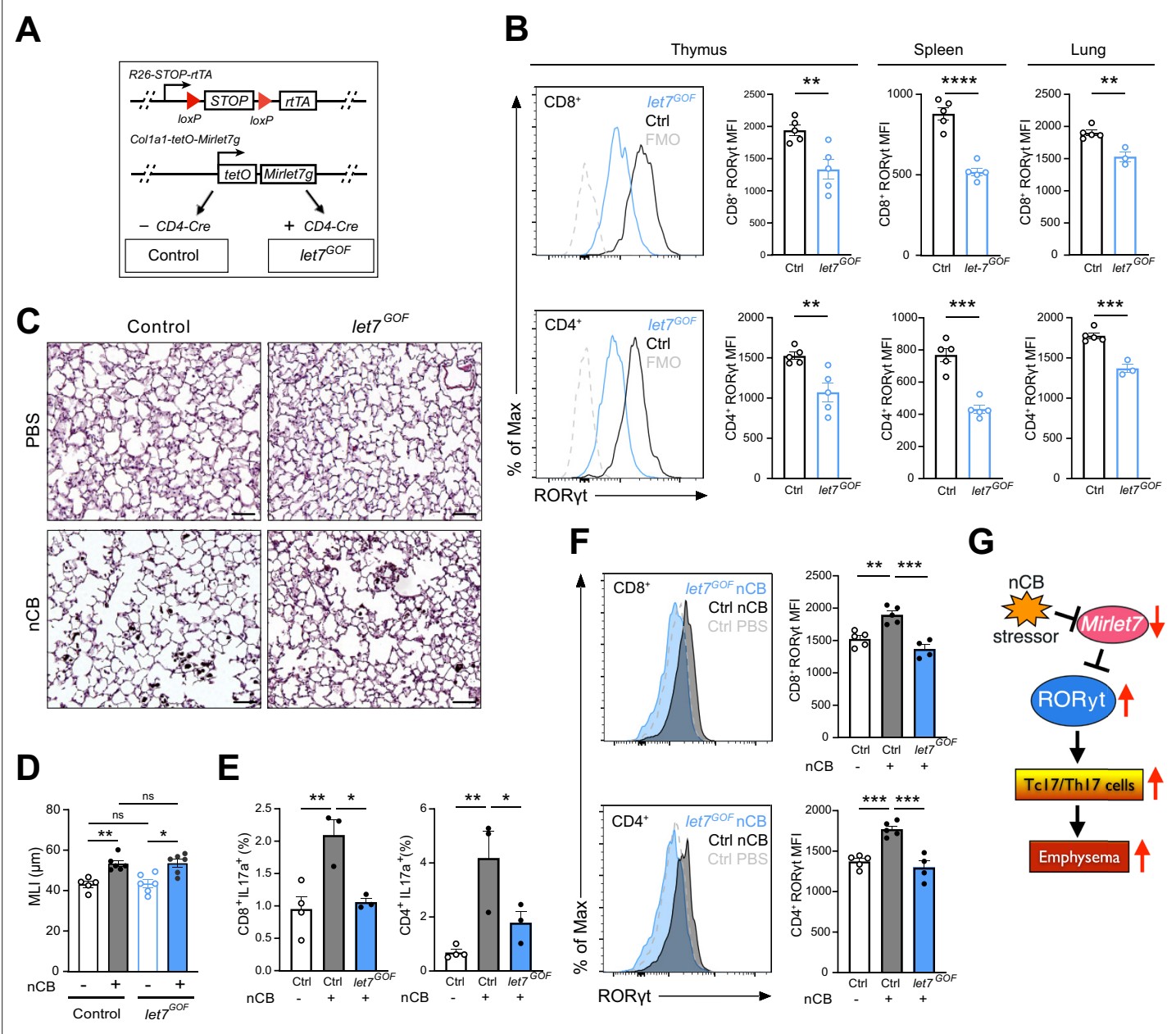

**Figure 6.** Enforced *Mirlet7g* expression in T cells restrains induction of RORγt and Tc17/Th17 inflammation in lungs of nanosized carbon black (nCB)-exposed mice. (**A**) Schematic outlining our T cell inducible *Mirlet7g* mouse model (*let7^GOF*). Flow analysis of RORγt expression in live, TCRβ+CD8+ or CD4+ T cells from (**B**) indicated mice in thymus, spleen, and lungs (*n* = 3–5 per group). (**C**) Control and *let7^GOF* mice were treated with PBS vehicle or nCB then analyzed. Representative hematoxylin and eosin (H&E)-stained lung sections from PBS- and nCB-exposed mice as indicated on each panel (×20 magnification; scale bars, 50 μm). (**D**) Mean linear intercept (MLI) measurements from indicated mice (*n* = 5–6 per group). Flow analysis of lungs gated on live TCRβ+ CD8+ or CD4+ cells for (**E**) IL-17a+ population frequency (*n* = 3–4 per group) or (**F**) RORγt expression by representative flow plot and mean fluorescent intensity (MFI) quantification (*n* = 4–5 per group). (**G**) Figure model of *Mirlet7*/RORγt axis in emphysema pathogenesis. Data are representative of two (**B**) or three (**C–F**) independent experiments and displayed as mean ± standard error of the mean (SEM) using Student's *t*-test (**B**) or two-way analysis of variance (ANOVA) with Tukey's multiple correction (**D–F**). *p < 0.05, **p < 0.01, ***p < 0.001, ****p < 0.0001.

The online version of this article includes the following source data for figure 6:

**Source data 1.** Source data for *Figure 6B, D–F*.

and elicitation of emphysema mice orchestrates the emergence and accumulation of Tc17 cells which may act in parallel with Th17 cells to promote tissue damage.

Prior research has shown the importance of both transcriptional and post-transcriptional regulatory control of RORγt expression in T cells (*Ciofani et al., 2012*; *Donate et al., 2013*; *Medvedev et al., 1997*). Altogether, our in vivo studies establish *Mirlet7* as a new important link associated with regulation of RORγt and lung Tc17 differentiation in COPD. Our data also showed that in vivo conditional genetic ablation of individual *Mirlet7* clusters in T cells stimulates a rise in RORγt protein expression in single-positive thymocytes and peripheral CD8⁺ and CD4⁺ T cells while enforced *Mirlet7g* activity leads to partial repression of RORγt in T cells. Despite these alterations in RORγt expression in our *Mirlet7* T cell LOF mice, the mice did not exhibit spontaneous gross phenotypes in thymus, spleens, or lungs at baseline nor did they exhibit changes in Tc17/Th17 subpopulations. This may be due to the subtle and modest expression thresholding of RORγt detected in mice and/or residual *Mirlet7* expression in T cells. On the other hand, and in agreement with our Tc17 and experimental emphysema data, we observed enhanced RORγt expression in lungs of *Mirlet7* T cell LOF after treatment with nCB. We corroborated the importance of *Mirlet7* activity in Tc17 differentiation of ex vivo cultured CD8⁺ T cells, as well as in the direct post-transcriptional control of RORγt, suggesting that this defect, is in part, direct and cell autonomous. We did not ascertain whether deletion of *Mirlet7a1/Mirlet7f1/Mirlet7d* cluster is an equally or more effective modulator of experimentally induced emphysema. Nonetheless, we predict that under different cell stress contexts, the functions of *Mirlet7* clusters do not fully overlap due to differential thresholding of mRNAs.

Prior studies have elucidated the relative and absolute quantities of individual *Mirlet7* family members in murine thymocytes and peripheral T cells which range from ~2% to 30% (*Pobezinskaya et al., 2019*; *Pobezinsky et al., 2015*). Moreover, the same group reported that all *Mirlet7* miRNAs are coordinately downregulation following antigen stimulation through the T cell receptor (*Wells et al., 2017*). Another recent study discerned a role for the lncRNA, *CCAT1* (*colon cancer-associated transcript 1*) as a molecular decoy or sponge in human bronchial epithelial cells which drives downregulation of *Mirlet7c* following CS extract exposure (*Lu et al., 2017*). Thus, it seems likely that complex synergistic transcriptional and post-transcriptional mechanisms contribute to downregulation of *Mirlet7* activity in emphysematous T cells.

It is also important to note that *Mirlet7* has been reported to exert potent effects by titrating the levels of multiple gene targets in mechanisms that contribute to Th17 inflammatory response and influence diverse set of processes including T cell activation, proliferation, differentiation, and cell homing (*Angelou et al., 2019*; *Beachy et al., 2012*; *Bronevetsky et al., 2016*; *Pobezinskaya et al., 2019*; *Pobezinsky et al., 2015*; *Wells et al., 2017*). A particular feature of these studies has been the utilization of *Lin28b* transgenic mice to block maturation and activity of entire *Mirlet7* family to promote a LOF function phenotype (*Angelou et al., 2019*; *Piskounova et al., 2011*; *Pobezinskaya et al., 2019*; *Wells et al., 2023*; *Zhu et al., 2011*). Furthermore, *Lin28b* was recently reported to also influence transcriptome-wide ribosome occupancy and global miRNA biogenesis (*Tan et al., 2019*). Thus, it is likely that *Lin28b* transgenic overexpression gives rise stronger phenotypes than we observed in our single cluster *Mirlet7* T cell LOF mice. Nonetheless, unbiased omics-based methods will be needed to determine if other gene targets beyond RORγt synergistically potentiate the in vivo Tc17 response and emphysema phenotype in context of deletion of *Mirlet7b/Mirlet7c2* cluster.

Tc17 cells play a major role in microbial infections, providing a potent anti-viral response (*Hamada et al., 2009*; *Yeh et al., 2010*), while viral infection has been an established factor in COPD exacerbations (*Hewitt et al., 2016*; *Wedzicha, 2004*). It will be interesting to determine whether loss of *Mirlet7* activity in the T cell compartment contributes to COPD disease susceptibility in the context of viral exposure. Our experiments with *Mirlet7g* GOF were partially successful in limiting the emergence of Tc17 and Th17 in nCB-elicited emphysema. The *Mirlet7g* GOF mice exhibited a reduction in RORγt expression levels and type 17 responses but were not protected from alveolar remodeling following nCB exposure. A potential limitation of this transgenic model is that it expresses only the *Mirlet7g* sequence which may render it less potent than the corresponding two mature forms transcribed from the *Mirlet7b/Mirlet7c2* cluster. Additional studies will be required to ascertain whether other interventions or alternative mouse models that enhance *Mirlet7* activity in T cells are successful in preventing or reversing COPD.

# Materials and methods

## Mice

Conditional knockout-ready floxed for either the *Mirlet7b/Mirlet7c2 and Mirlet7a1/Mirlet7f1/Mirlet7d* cluster, respectively, were generated using CRISPR gene editing in an isogenic C57BL/6 genetic background and were sequence verified for rigor. Mice were PCR genotyped from ear samples with primers flanking loxP sites (*Supplementary file 1*). Mice were crossed to *CD4-Cre* obtained from JAX (Stock 022071) and genotyped by PCR. The *R26-STOP-rtTA; Col1a1-tet0-Mirlet7g* (*rtTA-iLet7*) mice were obtained from JAX (Stocks 023912 and 05670) and also PCR genotyped with established JAX primers (*Belteki et al., 2005*; *Zhu et al., 2011*). Control *rtTA-iLet7* and *Mirlet7g* GOF mice were fed ad libitum with 200 mg/kg of doxycycline-containing chow (Bio-Serv S3888) at weaning age (*Belteki et al., 2005*; *Zhu et al., 2011*). Syngeneic littermates served as controls for all mouse experiments. All mice were bred in the transgenic animal facility at Baylor College of Medicine. All experimental protocols used in this study were approved by the Institutional Animal Care and Use Committee of Baylor College of Medicine animal protocol (AN-7389) and followed the National Research Council Guide for the Care and Use of Laboratory Animals.

## Human emphysema tissue samples and T cell isolation

Lung tissues were obtained from a total of 19 non-atopic current or former smokers with significant (>20 pack-years, one pack-year equals to smoking one pack of cigarettes per day each year) history of smoking who were recruited into studies from the chest or surgical clinics at Michael E. DeBakey Houston Veterans Affairs Medical Center hospitals (*Supplementary file 2*; *Shan et al., 2009*). Emphysema and non-emphysema control patients were diagnosed from CT scans according to the criteria recommended by the National Institutes of Health–World Health Organization workshop summary (*Pauwels et al., 2001*). Human single-cell suspensions were prepared from surgically resected lungs as previously described (*Yuan et al., 2019*; *Grumelli et al., 2004*). Briefly, fresh lung tissue was minced into 0.1 cm pieces in Petri dishes and treated with 2 mg/ml of collagenase D (Worthington) for 1 hr at 37°C. Digested lung tissue was filtered through a 40-μm cell strainer (BD Falcon) followed by red blood cell lysis using ACK lysis buffer (Sigma-Aldrich) for 3 min to yield a single-cell suspension. CD4$^+$ T cells were selected from resultant suspensions by labeling with bead conjugated anti-CD4 for enrichment by autoMACs (Miltenyi Biotec). Studies were approved by the Institutional Review Board at Baylor College of Medicine and informed consent was obtained from all patients.

## Human lung transcriptome data

A publicly available RNA-seq dataset from a Korean cohort GSE57148 was selected for the analysis (*Kim et al., 2015*). The raw FASTQ files of paired end reads representing the transcriptome of control and cases were retrieved from the GEO database at the National Centre for Biological Information (NCBI) through accession number GSE57148 and analyzed with R package for differential expression.

## CS exposure model of pulmonary emphysema

To promote emphysema, mice were exposed to CS using our custom designed whole-body inhalation system (*Morales-Mantilla et al., 2020*). In total, mice were exposed to four cigarettes (Marlboro 100's; Philip Morris USA) per day, 5 days a week, for 4 months as previously described (*Shan et al., 2012*).

## nCB exposure model of pulmonary emphysema

Nanosized particulate carbon black was prepared and administered as previously described (*You et al., 2015*; *Lu et al., 2015*). Dried nCB nanoparticles were resuspended in sterile PBS to a concentration of 10 mg/ml. Fifty μl of reconstituted nCB (0.5 mg) were intranasally delivered to deeply anesthetized mice on a schedule of three times a week for 4 weeks (total delivered dose of 6 mg). Lung histomorphometry and airway inflammation were assessed 4 weeks after the final nCB challenge. For histomorphometric analysis, mice lungs were fixed with 10% neutral-buffered formalin solution via a tracheal cannula at 25 cm $H_2O$ pressure followed by paraffin embedding and tissue sectioning and stained with H&E. MLI measurement of mouse lung morphometry was done as previously described (*Shan et al., 2014*; *Morales-Mantilla et al., 2020*). Briefly, this was done in a blinded fashion to mice genotypes from 10 randomly selected fields of lung parenchyma sections. Paralleled lines were placed

on serial lung sections and MLI was calculated by multiplying the length and the number of lines per field, divided by the number of intercepts (*Morales-Mantilla et al., 2020*).

BALF was collected by instilling and withdrawing 0.8 ml of sterile PBS twice through the trachea. Total and differential cell counts in the BALF were determined with the standard hemocytometer and HEMA3 staining (Biochemical Sciences Inc, Swedesboro, NJ) using 200 µl of BALF for cytospin slide preparation (*Morales-Mantilla et al., 2020*; *Lu et al., 2015*).

## Cell isolation from murine lung tissue

Mouse lung tissue was cut into 2 mm pieces and digested with collagenase type D (2 mg/ml; Worthington) and deoxyribonuclease (DNase) I (0.04 mg/ml; Roche) for 1 hr in a 37°C incubator. Single-cell suspensions from lung digest, spleen, and thymus were prepared by mincing through 40-µm cell strainers then washing and resuspension in complete RPMI media. Mouse lung and spleen single-cell suspensions were additionally overlaid on Lympholyte M cell separation media (Cedarlane) as indicated in the manufacturer's protocol to purify lymphocytes. For murine *Mirlet7* expression studies, lung single-cell suspensions were labeled with anti-CD4$^+$ or anti-CD8$^+$ magnetic beads and separated by autoMACS (Miltenyi Biotec), or CD4$^+$CD8$^+$ DP cells purified from thymus single-cell suspensions by flow-cytometric sorting on FACS Aria (BD Biosciences).

## In vitro polarization of CD8$^+$ T cells

CD8$^+$ naive T cells were isolated from spleen using Mojosort Mouse CD8 Naïve T cell isolation Kit (Biolegend) and adjusted to a concentration of $1.0 \times 10^6$ cells/ml. Purified cells were activated with plate-bound anti-CD3 (1.5 µg/mL) and complete RPMI media containing anti-CD28 (1.5 µg/ml) and β-mercaptoethanol (50 nM) for Tc0 polarization, or further supplemented with Tc1 [IL-2 (10 ng/ml)] or Tc17 [TGFβ (2 ng/ml), IL-6 (20 ng/ml), anti-IFNγ (10 µg/ml), IL-23 (20 ng/ml), and IL-1β (5 ng/ml)] polarization conditions for 72 hr (*Flores-Santibáñez et al., 2018*).

## ELISA

Supernatant was collected from in vitro polarized murine CD8$^+$ T cells and centrifuged to remove cellular debris (*Lu et al., 2015*). Cytokine levels of IL-17A and IFNγ were quantified from collected supernatant using Mouse IL-17A Uncoated ELISA and Mouse IFN gamma Uncoated ELISA (Invitrogen) Kits, respectively, per the manufacturer's instructions with colorimetric analysis by the Varioskan LUX microplate reader (Thermo Fisher).

## Flow cytometric analysis

Cells used for in vitro or in vivo cytokine analysis were stimulated with PMA (20 ng/ml; Sigma-Aldrich), Ionomycin (1 µg/ml; Sigma-Aldrich), and Brefeldin A (2 µg/ml; Sigma-Aldrich) for 4 hr prior to flow staining (*Lu et al., 2015*). For intracellular staining, cells were fixed and permeabilized using the Mouse FOXP3 Buffer Set (BD) per the manufacturer's protocol. The fluorophore-conjugated antibodies used in this study were as follows: Live/Dead Fix Blue (Invitrogen), CD3 PerCPCy5.5 (Biolegend), TCRb PE/Cy7 (Biolegend), CD4 PB (Biolegend), CD4 AF700 (Biolegend), CD8 BV650 (Biolegend), CD25 BV421 (Biolegend), FOXP3 AF488 (Biolegend), ROR gamma T PE (Invitrogen), TCF1 AF647 (Cell Signaling Technologies), TCF1 PE (Biolegend), IFNγ AF647 (Biolegend), IL17A FITC (Biolegend), and IL17A PE (ebioscience). Samples were analyzed using BD LSR II flow cytometer (BD Biosciences) and FlowJo software (TreeStar).

## RNA isolation and quantitative RT-PCR

RNA was isolated using miRNeasy (QIAGEN) or RNeasy Mini Kit (QIAGEN) in conjunction with the RNase-Free DNase (QIAGEN) according to the manufacturer's instructions. cDNA of miRNAs and mRNAs was synthesized using TaqMan Advanced miRNA cDNA Synthesis Kit (Thermo Fisher) and High-Capacity cDNA Reverse Transcription Kit Real-Time PCR system (Applied Biosystems). 18S and snoRNA-202 were used to normalize mRNA and miRNA expression, respectively (*Lu et al., 2015*). Quantitative RT-PCR data were acquired on 7500 Real-Time PCR System or StepOne Real-Time PCR System (Applied Biosystems) with the following TaqMan probes: *hsa-let-7a-5p* [478575_mir], *hsa-let-7b-5p* [478576_mir], *hsa-let-7d* [478439_mir], *hsa-let-7f* [478578_mir], *pri-let7a1/f1/d* [44411114, arfvmhy], *pri-let7b/c2* [4441114, areptx2], *Mmp9* [Mm00442991], and *Mmp12* [Mm00500554].

## Luciferase reporter assays

Genomic fragment containing the murine *Rorc* 3′UTR was cloned into psiCHECK2 luciferase reporter plasmid (Promega). This construct was also used to generate the *Mirlet7* 'seed' deletion mutant derivative using the QuikChange Multi Site Mutagenesis Kit (catalog 200514-5, Stratagene). 3T3 mouse embryonic fibroblasts were transfected using Oligofectamine (Invitrogen) with 100 ng of psiCheck-2 plasmid containing wild-type or mutant 3′UTR, along with the miRNA control or *Mirlet7b* duplex (Dharmacon) at a final concentration of 6 nM (*Gurha et al., 2012*; *Supplementary file 1*). Reporter activity was detected with the Dual-Luciferase Reporter Assay System (Promega).

## Statistical analysis

Statistical analyses were performed using GraphPad Prism 10.0.1 software. Statistical comparison between groups was performed using the unpaired Student's *t*-test, two-way analysis of variance with Tukey's or Sidak's correction, and Mann–Whitney test when indicated. A p-value less than 0.05 was considered statistically significant; ns indicates not significant. Statistical significance values were set as $*p < 0.05$, $**p < 0.01$, $***p < 0.001$, and $****p < 0.0001$. Data are presented as means ± standard error of the mean. p-value and *n* can be found in the main and supplementary figure legends.

## Acknowledgements

We thank Jason Heaney and Denise Lanza at BCM Genetically Engineered Rodents Core funded in part by NIH P30 CA125123; Patricia Castro at Tissue Acquisition and Pathology Core funded in part by P30 CA125123; and Joel M Sederstrom at the BCM and Cell Sorting Core with funding from the CPRIT Core Facility Support Award (CPRIT-RP180672) and NIH (CA125123 and RR024574). This work was supported by grants from the NHLBI (R01HL140398 to AR), the Gilson Longenbaugh Foundation (to AR), and NIEHS (T32 ES027801 to PE).

---

## Additional information

### Funding

| Funder | Grant reference number | Author |
| --- | --- | --- |
| National Heart, Lung, and Blood Institute | R01HL140398 | Antony Rodriguez |
| National Institute of Environmental Health Sciences | T32 ES027801 | Phillip A Erice |
| Gilson Longenbaugh Foundation | | Antony Rodriguez |

The funders had no role in study design, data collection, and interpretation, or the decision to submit the work for publication.

### Author contributions

Phillip A Erice, Conceptualization, Data curation, Formal analysis, Supervision, Validation, Investigation, Visualization, Methodology, Writing – original draft, Writing – review and editing; Xinyan Huang, Conceptualization, Data curation, Formal analysis, Supervision, Validation, Investigation, Methodology; Matthew J Seasock, Hui-Ying Tung, Data curation, Formal analysis, Validation, Investigation, Methodology; Matthew J Robertson, Resources, Data curation, Software, Formal analysis, Investigation, Visualization, Methodology; Melissa A Perez-Negron, Shivani L Lotlikar, Formal analysis, Methodology; David B Corry, Conceptualization, Resources, Investigation, Methodology; Farrah Kheradmand, Conceptualization, Resources, Formal analysis, Investigation, Methodology; Antony Rodriguez, Conceptualization, Resources, Data curation, Formal analysis, Supervision, Funding acquisition, Validation, Investigation, Visualization, Methodology, Writing – original draft, Project administration, Writing – review and editing

## Author ORCIDs

Matthew J Seasock  https://orcid.org/0000-0002-5940-5913
Antony Rodriguez  http://orcid.org/0000-0001-7184-9413

## Ethics

This study was performed in strict accordance with the recommendations in the Guide for the Care and Use of Laboratory Animals of the National Institutes of Health. All of the animals were handled according to approved Institutional Animal Care and Use Committee (IACUC) protocol (AN-7398) at Baylor College of Medicine.

The original design, eligibility and results of Human Subjects samples was published (Shan et al., 2009). Patients were recruited into the chest or surgical clinics at Michael E. DeBakey Houston Veterans Affairs Medical Center hospitals. All studies were approved by the Institutional Review Board at Baylor College of Medicine and informed consent was obtained from all patients.

Reviewer #1 (Public Review): https://doi.org/10.7554/eLife.92879.3.sa1
Reviewer #2 (Public Review): https://doi.org/10.7554/eLife.92879.3.sa2
Author response https://doi.org/10.7554/eLife.92879.3.sa3

# Additional files

## Supplementary files

• Supplementary file 1. Genotyping primers and duplexes for luciferase assay. Primer names and sequences are indicated above.

• Supplementary file 2. Demographics of subjects by emphysema severity. Mean ± standard deviation is shown for age, $FEV_1$ %, $FEV_1/FVC$ %. Abbreviations: $FEV_1$ (forced expiratory volume in 1 s). $FEV_1/FVC$ (forced expiratory volume/forced vital capacity).

• MDAR checklist

## Data availability

Source data files for main figures are provided in the manuscript.

The following previously published dataset was used:

| Author(s) | Year | Dataset title | Dataset URL | Database and Identifier |
|---|---|---|---|---|
| Kim WJ, Lim JH, Kim WJ, Kim J, Lee JS, Oh Y, Lee SD | 2015 | Comprehensive Analysis of Transcriptome Sequencing Data in the Lung Tissues of COPD Subjects | https://www.ncbi.nlm.nih.gov/geo/query/acc.cgi?acc=GSE57148 | NCBI Gene Expression Omnibus, GSE57148 |

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
