## [Editor Report · eLife assessment]

This **important** study indicates a significant role for individual *let-7* miRNA clusters in regulating generation of Tc17 CD8 cells and emphysema severity in a mouse model. The authors provide **convincing** evidence for let-7-mediated repression of the transcription factor RORgt and consequent modulation of IL-17-producing CD8 T cells, with correlated data from human emphysema material, though some of the effective *let-7* clusters remain to be tested for the ability to modulate disease. The findings, which substantially advance the understanding of roles that *let-7* miRNA clusters play in modulating both T cell responses and emphysematous lung disease, will be of interest to T cell and lung disease researchers.

---

## [Referee Report · Reviewer #1 (Public Review)]

Summary:

Inflammatory T cells have been recognized to play an important role in human COPD lung tissue and animal models of emphysema. The authors have previously identified that Th17 cells regulate chronic inflammatory diseases, including in mice exposed to smoke or nanoparticulate carbon black (nCB). Here, the authors interrogate the role of Tc17 cells using similar mouse models. Investigating let-7 miRNA, which induces antigen-presenting cell activation and T cell-mediated Th17a inflammation, they show that the master regulator of Tc17/Th17 differentiation, RAR-related orphan receptor gamma t (RORγt), is a direct target of let-7 miRNA in T cells. Because RORγt expression is elevated in COPD patients and in mouse models of COPD, the authors generate a Let-7 overexpressing mouse in T cells and reduce RORγt expression and Th17 and Tc17 cell recruitment in nCB-exposed mice.

Strengths:

The authors use a previously published RNA-seq dataset (GSE57148) from lungs of control and COPD subjects to explore the involvement of Let-7 in emphysema. They further evaluate Let-7a expression by qPCR in lung tissue samples of smokers with emphysema and non-emphysema controls. Moreover, expression of Let-7a, Let-7b, Let-7d, and Let-7f in purified CD4+ T cells were inversely correlated with emphysema severity lungs. Similar findings were found in their mouse models (CS or nCB) in both lung tissue and isolated lung CD4+ and CD8+ T cells, with reduced let-7afd and let-7bc2 expression.

Using mice harboring a conditional deletion of the let-7bc2 cluster in all T cells (let-7bc2LOF) derived from the CD4+CD8+ double-positive stage, the authors show enhanced emphysema in nCB- or CS-exposed mice with enhanced recruitment of macrophages and neutrophils to the lung. While CD8+IL17a+ Tc17 cells and CD4+ IL17a+ Th17 cells were increased in nCB-exposed control animals, only let-7bc2LOF mice showed an increase in CD8+IL17a+ Tc17 cells. Further, unexposed let-7bc2LOF and let-7afdLOF mice expressed greater RORγt expression in both CD8+ and CD4+ T cells.

Generating a let-7 gain of function mouse with overexpression of let-7g in thymic double-positive-derived T cells, protein levels of RORγt were suppressed in CD8+ and CD4+ T cells of let-7GOF mice relative to controls. Let-7GOF mice treated with nCB showed similar lung alveolar distension as controls suggesting that increased let-7 expression does not protect the lung from emphysema. However, let-7GOF mice showed reduced lung Tc17 and Th17 cell populations and were resistant to the induction of RORγt after nCB exposure.

Weaknesses:

Limited data is shown on the let-7afdLOF mice. Does this mouse respond similarly to nCB as the let-7bc2LOF.

Because the authors validate their findings from a previously published RNA-seq dataset in subjects with and without emphysema, the authors should include patient demographics from the data presented in Figure 1C-D.

To validate their mouse models, the absence of Let-7 or enhanced Let-7 expression needs to be shown in isolated T cells from exposed mice.

In Figure 3, the authors are missing the unexposed let-7bc2LOF group from all panels. This is again an issue in Figure 6 with the let-7GOF.

Because the GOF mouse enhances Let-7g within T cells, the importance of Let-7g should be determined in human subjects. Why did the authors choose to overexpress Let-7g, the rationale is not clear.

The purity of the CD4+ and CD8+ T cells is not shown and the full gating strategy should be included.

The authors indicate that Tc17 and Th17 T cells were reduced in the GOF mouse, it remains unclear if macrophage or neutrophil recruitment is altered in GOF mice.

---

## [Referee Report · Reviewer #2 (Public Review)]

Summary:

This valuable study characterizes the requirement for individual let-7 clusters to limit the generation of IL-17 producing CD8 T cells and the severity of emphysema in mouse models. Mature let-7 family miRNAs originate from multiple loci, several of which have been reported and/or are reported here to be downregulated in emphysematous lung tissue and/or lung T cells. The results provided are convincing but incomplete, as the let-7 cluster with the most convincing effects on T cell cytokine production is not tested for effects on disease pathogenesis.

Let-7 family miRNAs are largely redundant in function and originate from multiple genomic loci ("clusters"). Erice et al demonstrate that two individual clusters (let7afd and let7bc2) in mice regulate the generation of IL-17 producing CD8 T cells in vitro and in vivo in a model of emphysema. These cells also express higher levels of the IL-17-inducing transcription factor RORgt, encoded by Rorc, which the authors demonstrate to be a direct target of let-7. Since multiple let-7 family miRNAs are downregulated in T cells and lung tissue in emphysema, these data support a model in which reduced let-7 allows increased IL-17 production by T cells, contributing to disease pathogenesis.

Strengths:

The inclusion of miRNA and pri-miRNA expression data from sorted human lung T cells as well as mouse T cells from an emphysema model is a strength.

The study includes complementary loss of function and gain of function experimental systems to test the effect of altered let-7 function, though it should be noted that these involved different let-7 family members and did not yield simple, complementary results for all experimental outcomes.

The most important finding is that deletion of just one let-7 cluster ("Let7bc2") is sufficient to exacerbate emphysema in the nCB and CS models.

Weaknesses:

The human miRNA expression data that motivate functional analyses used sorted CD4+ T cells. The authors note that prior work on let-7 showed that it regulates Th17 (CD4) responses, yet this study's functional analyses are all focused on Tc17 (CD8) T cells. Data in this paper show that Tc17 cells are far less numerous than Th17 cells in the nCB and CS models of emphysema.

Compared with Let7bc2 deletion, Let7afd deletion had a much larger effect on IL17 production by CD8 T cells in vitro, and it also had a larger effect on RORgt expression in untreated mice in vivo, especially in the lung. In the revised manuscript, the authors show that let7afdLOF mice have normal numbers of CD4 and CD8 T cells in the thymus and peripheral lymphoid organs and do not exhibit lung histopathology or inflammatory changes at baseline at least up to 6 months of age. As such, they are set up perfectly to test the requirement for Let7afd in the nCB and/or CS models. These experiments would add strength to the core novelty of this work - demonstration of the functional importance of individual let-7 clusters.

The authors could do more to explain the complexity of the let7 miRNA family and the genomic clusters examined in this study. In particular, it would help to know the relationship between mouse Let7bc2 and corresponding human Let7 clusters. It would also be very helpful to know the relative expression of each mature let-7 family member in Tc17 cells. Are mature miRNAs derived from the Let7afd cluster more or less abundant?

The provided evidence for the effect of Let7GOF has an important caveat that came to light during review. Let7g overexpression caused a marked reduction in Rorgt expression in T cells at baseline and in the setting of nCB challenge, and it reduced the frequency of IL17+ producing CD8 T cells in the lung to baseline levels. Yet there was no change in the MLI measurement of histopathology. However, the responses in the experiment shown in Fig. 6C-D are quite muted compared to those shown in Figure 2. In the response to reviewers, the authors speculate that an anti-inflammatory of doxycycline, required for induction of Let7g in this model, "could account for the differences in the magnitude of emphysemic response".

Although RORgt is a great candidate to have direct effects on IL-17 expression, the mechanistic understanding of let-7 action on T cell differentiation and cytokine production is limited to this single target. As noted in the discussion, others have identified cytokine receptor targets that may play a role, but it is also likely others among the many targets of let-7 also contribute.

---

## [Author Response]

The following is the authors’ response to the original reviews.

We would like to thank the editors and reviewers for providing feedback and suggestions for our manuscript.

In response to reviewers comments we changed several main Figures and added new tables and supplementary figures. We also made edits to the Discussion.

**Reviewer #1 (Public Review):**
Weaknesses:Limited data is shown on the let-7afdLOF mice. Does this mouse respond similarly to nCB as the let-7bc2LOF.

In the revised manuscript, we have added a baseline lung phenotypic assessment for the let-7afdLOF mice up to 6-months of age within Figure 4-figure supplement 1. The data supports our original statement and observation that let-7afdLOF mice do not exhibit lung pathology, inflammation, or changes in T cell subsets at baseline. Our view is that current manuscript addresses the importance of let-7bc2-cluster in experimental emphysema and the let-7afd-cluster mice is used to validate Rorc as a direct target of let-7. In the future, new grant funding will make it possible to ascertain whether absence of the let-7afd-cluster also sensitizes mice to experimentally induced emphysema.

Because the authors validate their findings from a previously published RNA-seq dataset in subjects with and without emphysema, the authors should include patient demographics from the data presented in Figure 1C-D.

We thank the reviewers for their recommendation. In address of this, the revised manuscript contains a new Supplementary Table 1 with the human subject demographic information that corresponds with Figure 1D.

To validate their mouse models, the absence of Let-7 or enhanced Let-7 expression needs to be shown in isolated T cells from exposed mice.

In the case of let-7bc2-cluster, we have included Figure 2-figure supplement 2 which shows pri-let7bc2 expression assessed by qPCR from selected CD8+ lung T cells of control and let-7bc2LOF mice exposed to PBS vehicle or nCB. The let-7g GOF model used in our studies has been validated for the induction of let-7g in thymic and peripheral T cells and elicitation of gain-of-function phenotypes (Pobezinskaya et al. 2019; Angelou et al. 2020; Wells et al. 2023).

In Figure 3, the authors are missing the unexposed let-7bc2LOF group from all panels.

We emphasize that our exhaustive characterization of control and let-7bc2LOF mice in absence of challenge showed no phenotype. The baseline data was collectively shown in Figure 2-figure supplement 1.

Why did the authors choose to overexpress Let-7g, the rational is not clear?

We concur that ideal GOF experiments can be carried out with let-7b or let-7c. Unfortunately, let-7b/c2 transgenic mice are not currently available, so we elected to use the well characterized let-7g T cell GOF mouse model (Pobezinskaya et al. 2019; Angelou et al. 2020; Wells et al. 2023). Furthermore, it is worth noting that the binding/seed sequence of let-7g is identical to let-7a/b/c and other members. Nonetheless, we have edited our Discussion section to reflect this as a potential caveat that can confound the utilization of this let-7GOF mouse model.

The purity of the CD4+ and CD8+ T cells is not shown and the full gating strategy should be included.

In the revision, we included the flow gating strategy and display the representative population with purities in Supplementary Figure 1 of the revised manuscript.

**Reviewer #2 (Public Review):**
Weaknesses:The functional analyses are unusually focused on IL-17 producing CD8 T cells, but it is not made clear whether these cells are an important player in emphysema pathogenesis in the nCB and CS models. The data shown reveal that they are far less numerous than IL-17-producing CD4 T cells. It is also notable that the Figure 1 expression data from human subjects used sorted CD4+ T cells. And as the author mentioned, prior work on let-7 showed that it regulated Th17 (CD4) responses.

As we showed that the let-7bc2LOF had enhanced the Tc17 cell population without any significant impact on Th17 cells, we elected to focus our analysis on this population. Furthermore, the connection of let-7 with the generation of a Tc17 inflammatory response is a novel finding, which so far remained unappreciated in the field and instigates new lines of inquiry.

Compared with Let7bc2 deletion, Let7afd deletion had a much larger effect on IL17 production by CD8 T cells in vitro, and it also had a larger effect on RORgt expression in untreated mice in vivo, especially in the lung. It would be valuable to more thoroughly characterize the let7afd mice. RORgt expression should be shown in the in vitro assays. In the results, the authors state that let7afdLOF mice "did not exhibit lung histopathology nor inflammatory changes" up to 6 months of age. Similarly, it is stated in the conclusion that "the let-7afdLOF mice ... did not exhibit changes in Tc17/Th17 subpopulations" in vivo. All these data should be shown, and if no baseline changes are apparent, then I also recommend challenging these mice with nCB and/or cigarette smoke.

We concur that additional phenotypic characterization on the let-7afdLOF mice will contribute valuable information in the future. Reviewer 1 had a similar comment. As described above in response to Reviewer 1, we added comprehensive phenotypic analysis of let-7afdLOF mice within Figure 4-figure supplement 1 in the revised manuscript. The new data indicates that there is no overt lung pathology in the let-7afdLOF mice despite the subtle induction of RORγt expression in T cells. Furthermore, we have now included flow cytometric analysis of RORγt expression from in vitro polarized Tc0 and Tc17 cells from let-7afdLOF mice within revised Figure 5H.

This brings up the larger issue of redundancy among the let-7 family members and genomic clusters. This should be discussed, including some explanation of the relative expression of each mature family member in T cells, and how that maps to the clusters studied here (and those that were not investigated). It would also be helpful to explain the relationship between mouse Let7bc2 and human Let7a3b, since Let7bc2 is the primary focus of emphysema experiments in this manuscript. This is especially important because the study of individual let-7 clusters is the core novelty of this body of work, as described in the first paragraph of the discussion. The regulation of let-7 expression has been reported before and its functional role has been investigated with a variety of tools.

We appreciate the interest and suggestion to expand the discussion on the let-7 family and their expression regulation. To address these points, we included additional references and expanded the Discussion section of the revised manuscript.

Let7g overexpression caused a marked reduction in Rorgt expression in T cells at baseline and in the setting of nCB challenge, and it reduced the frequency of IL17+ producing CD8 T cells in the lung to baseline levels. Yet there was no change in the MLI measurement of histopathology. Is this a robust result? The responses in the experiment shown in Fig. 6C-D are quite muted compared to those shown in Figure 2. The latter also shows a larger number of replicates, and it is unclear whether the data in 6D include measurement from all of the mice tested (e.g. pooled from 2 small experiments) or only mice from one experiment.

We appreciate the reviewer inquiry into the data presented in Figure 6C-D. The data is representative of a single experiment and the number of experiments has been added to the revised Figure 6 legend. We note that all let-7GOF and associated control mice in Figure 6 are exposed to doxycycline as part of the let7g induction model, whereas mice in Figure 2 are not. It has been previously reported that doxycycline, a member of the tetracycline family of molecules, has anti-inflammatory properties (Di Caprio et al. 2015), which we speculate could account for the differences in the magnitude of emphysemic response.

**Reviewer #3 (Public Review):**
Weaknesses:The authors show no change in frequencies of Treg cells in let-7bc2LOF mice exposed to nCB. Do these Treg cells also express higher levels of RORgt and IL-17? The major question that was not addressed in this study is how let-7 expression is regulated in emphysema. The other recommendation is that the authors include the sequences of the let-7 mimic oligos used in the luciferase assay.

We did not have the opportunity to address whether RORγt is in fact also upregulated in Treg cells. It remains unclear what upstream mechanisms drive the downregulation of the let-7 clusters in T cells with exposure to smoke/nCB. However, we agree that this an important question and we therefore updated the Discussion section of manuscript by including several citations that could explain how let-7 clusters become repressed in a coordinated fashion. Regarding the last point, the sequence of the duplex used in luciferase assay corresponds to the canonical mature let-7b in NCBI and has been added to Supplementary Table 3.

**Reviewer #2 (Recommendations For The Authors):**
The authors state that "Recent evidence suggests the let-7 family is downregulated in patients with COPD, however, how they cause emphysema remains unclear." This should be reworded. Its downregulation in disease does not necessarily indicate that let-7 causes emphysema. Also, recommend rewording "Overall, our findings shed light on the let-7/RORγt axis as a braking and driving regulatory circuit in the generation of Tc17 cells..." What does it mean to be a "braking and driving" circuit? These terms seem contradictory.

We recognize that the sentences were not phrased clearly. We have rephrased these statements as “Recent evidence suggests the let-7 miRNA family is downregulated in patients with COPD, however, whether this repression conveys a functional consequence in emphysema pathology has not been elucidated.” and “Overall, our findings shed light on the let-7/RORγt axis with let-7 acting as a molecular brake in the generation of Tc17 cells…”

Experimental details are needed for the human miRNA expression studies. Too little information is provided in the methods section, and the article cited there (Yuan et al 2020) is not listed in the bibliography.

We expanded the Materials and Methods section for the collection, isolation, and qPCR analysis of human subject lung T cells. We have corrected the bibliography and added the missing citation.

The claim of novelty for miRNA-mediated silencing of Rorc in the discussion section is unnecessary and incorrect (https://pubmed.ncbi.nlm.nih.gov/23359619).

Thank you for bringing the publication to our attention. Close inspection of this publication indicates that the authors did not experimentally validate Rorc as a direct target of let-7 itself. Plus the work was limited to immortalized in vitro cell cultures. We amended the sentence in the Discussion section highlighting the novelty of our findings which is the demonstration of Rorc as an in vivo target of let-7 in T cells.

Citations

Angelou, Constance C., Alexandria C. Wells, Jyothi Vijayaraghavan, Carey E. Dougan, Rebecca Lawlor, Elizabeth Iverson, Vanja Lazarevic, et al. 2020. “Differentiation of Pathogenic Th17 Cells Is Negatively Regulated by Let-7 MicroRNAs in a Mouse Model of Multiple Sclerosis.” Frontiers in Immunology 10: 3125. https://doi.org/10.3389/fimmu.2019.03125.

Di Caprio, Roberta, Serena Lembo, Luisa Di Costanzo, Anna Balato, and Giuseppe Monfrecola. 2015. “Anti-Inflammatory Properties of Low and High Doxycycline Doses: An in Vitro Study.” Mediators of Inflammation 2015: 329418. https://doi.org/10.1155/2015/329418.

Pobezinskaya, Elena L., Alexandria C. Wells, Constance C. Angelou, Eric Fagerberg, Esengul Aral, Elizabeth Iverson, Motoko Y. Kimura, and Leonid A. Pobezinsky. 2019. “Survival of Naïve T Cells Requires the Expression of Let-7 miRNAs.” Frontiers in Immunology 10 (May).https://doi.org/10.3389/fimmu.2019.00955.

Wells, Alexandria C., Kaito A. Hioki, Constance C. Angelou, Adam C. Lynch, Xueting Liang, Daniel J. Ryan, Iris Thesmar, et al. 2023. “Let-7 Enhances Murine Anti-Tumor CD8 T Cell Responses by Promoting Memory and Antagonizing Terminal Differentiation.” Nature Communications 14 (1): 5585. https://doi.org/10.1038/s41467-023-40959-7.